# ADVERSARIAL DIVERSITY IN HANABI

**Brandon Cui** [*†]
MosaicML

**Andrei Lupu**[*]
Meta AI & FLAIR, University of Oxford

**Samuel Sokota**[†]
Carnegie Mellon University

**Hengyuan Hu**[†]
Stanford University

**David J Wu**
Meta AI

**Jakob N. Foerster** [†]
FLAIR, University of Oxford

## ABSTRACT

Many Dec-POMDPs admit a qualitatively diverse set of "reasonable" joint policies, where reasonableness is indicated by symmetry equivariance, non-sabotaging behaviour and the graceful degradation of performance when paired with ad-hoc partners. Some of the work in diversity literature is concerned with generating these policies. Unfortunately, existing methods fail to produce teams of agents that are simultaneously diverse, high performing, and reasonable. In this work, we propose a novel approach, *adversarial diversity* (ADVERSITY), which is designed for turn-based Dec-POMDPs with public actions. ADVERSITY relies on off-belief learning to encourage reasonableness and skill, and on "repulsive" fictitious transitions to encourage diversity. We use this approach to generate new agents with distinct but reasonable play styles for the card game Hanabi and open-source our agents to be used for future research on (ad-hoc) coordination.[1]

## 1 INTRODUCTION

A key objective of cooperative multi-agent reinforcement learning (MARL) is to produce agents capable of coordinating with novel partners, including other artificial agents and ultimately humans. In order to make progress on this objective, a number of works have focused on the general challenge of *ad-hoc team play*, which is to create autonomous agents able to "*efficiently and robustly collaborate with previously unknown teammates on tasks to which they are all individually capable of contributing as team members*" (Stone et al., 2010). To evaluate such agents, many works on ad-hoc coordination rely on evaluation setups similar to the one proposed by Stone et al. (2010), pairing the agents at test time with partners sampled from a pre-determined pool.

The value of such evaluations depends on the size and quality of the pool of partners. A pool that is too small or too homogeneous may not be representative of all possible play-styles, and provide an inaccurate evaluation of the coordination capabilities of an agent. For this reason, previous works in coordination have relied on various approaches to generate a diverse pool of partners.

A first approach is to handcraft policies, either directly or by shaping the reward at train-time (Albrecht, 2015; Barrett et al., 2017; Zand et al., 2022), but it requires domain knowledge and scales poorly. Another is to train a population with varying hyperparameters or by deploying multiple RL algorithms on the same task (Nekoei et al., 2021; Zand et al., 2022; Albrecht, 2015). The diversity achieved this way is unclear, since it is a byproduct of the variability of the algorithms used rather than being actively optimized for. Yet other works augment training with a diversity loss (Lupu et al., 2021) or save multiple checkpoints (Strouse et al., 2021) but often do not report the level of diversity achieved. Measures of diversity based on policy similarity struggle in settings where not all different actions result in "meaningfully" different outcomes[2]. Furthermore, the number of possible *trajectories* is often so large that it becomes easy to maximize diversity objectives without learning qualitatively different policies – imagine a humanoid robot that wiggles a finger at any sin-

---

[*]Equal contribution. Correspondence at `brandon@mosaicml.com` and `alupu@meta.com`
[†]Work done while at Meta AI.
[1]`https://github.com/facebookresearch/off-belief-learning`
[2]While "meaningfully different" is environment dependent, we elaborate on what we mean in Section 4

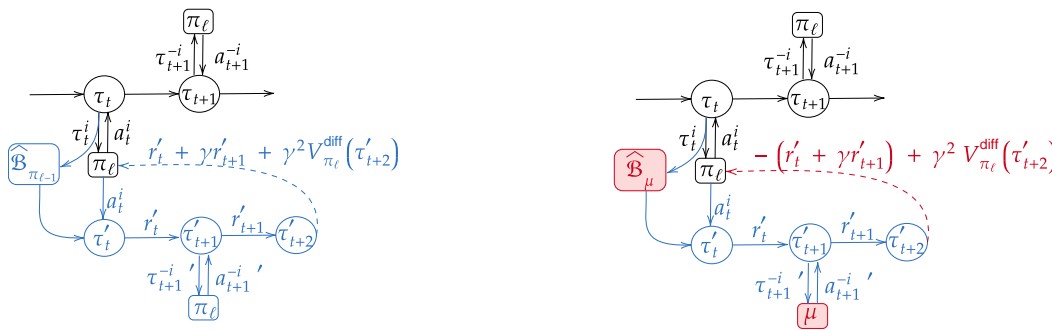

(a) Standard OBL transition          (b) Repulsive OBL transition

Figure 1: Standard (a) and repulsive (b) OBL transitions when training $\pi_\ell$ for $n = 2$ steps. ADVERSITY trains policy $\pi_\ell$ on (b) with probability $\lambda$, and on (a) otherwise. Differences between the two are in red.

gle time step in the episode rather than learning different walking styles. This is particularly true in multi-agent settings, where the number of trajectories is exponential in the number of agents.

To avoid such pitfalls, another approach, followed by Charakorn et al., is to require distinct policies to be *incompatible* by training them to obtain a low score when paired in mixed teams. In Section 7, we show that this results in policies that simply identify whether they are playing in self-play (SP, with themselves) or in cross play (XP), with another agent. In the latter case, they purposely "*sabotage*" the game by selecting actions that minimize return, such as playing unplayable cards in the card game Hanabi. Adapting to such policies in an ad-hoc pool is a non-goal, since they do not represent meaningfully different policies but rather *actively poor and adversarial game play*.

This is in line with previous findings that partners trained with SP rely on arbitrary conventions and symmetry breaking, making collaboration with them difficult (Hu et al., 2020; 2021b; Lupu et al., 2021). As such, producing strong and *meaningfully diverse* policies in Dec-POMDPs remains an important unsolved problem.

We address this problem in turn-based settings with public actions by introducing adversarial diversity (ADVERSITY), a policy training method which, given a *repulser agent*, produces an "adversary" whose conventions are fundamentally incompatible with those of the repulser. The key insight of ADVERSITY is that it prevents the adversary from *identifying* whether it is currently in SP or playing with the repulser agent by *randomizing* between the two at every time step. In other words, even if the adversary is in an action-observation history (AOH) that is *incompatible* with the repulser agent, the adversary is paired with the repulser agent with a fixed probability $\lambda$, in which case the next reward is inverted. Likewise, with probability $1 - \lambda$ the adversary is instead paired with itself.

Crucially, the choice of the current partner determines not only the sign of the reward (positive or negative) and the partners' action, but also how the entire AOH thus far is *interpreted*. Here, we build on top of the *fictitious transition* mechanism from off-belief learning (Hu et al., 2021b, OBL) and use the belief model of the repulser policy on the corresponding *repulsive transition*. In a nutshell, if the adversary is currently paired with the repulser policy, the transition is sampled from a *belief distribution* that assumes the repulser policy took all actions thus far.

When the adversary is paired with itself rather than the repulser, we must avoid the feedback loop between induced beliefs and future actions, which would allow the adversary to form *arbitrary conventions*. Thus, we train in a hierarchy: we start with the *grounded belief*, like in OBL, and at each level $\ell$ we compute the vanilla transitions using the belief model of the level below, $\ell - 1$. The adversary is trained to maximize a "*difference value function*", which estimates the *forward looking discounted difference* between adversarial and vanilla transitions under their corresponding beliefs.

For the first time, ADVERSITY enables us to produce a number of high performing, diverse, and symmetry-invariant policies for the challenging collaborative card game Hanabi.

## 2 RELATED WORK

Stone et al. (2010) and Bowling & McCracken (2005) were among the first to formulate the ad-hoc teamwork ("impromptu team play") setting, requiring autonomous agents to collaborate with novel teammates. Works in the literature have often taken a type-based approach, where potential partners are grouped in a number of possible types, which must be identified at test time. Different types (or classes) of polices have notably been generated through genetic algorithms (Albrecht et al., 2015) to

induce diversity. However, these methods usually require hand-coded heuristics and are difficult to scale to complex, high dimensional environments that we consider in this paper.

Other works promote diversity in multi-agent RL (MARL) to robustify an agent by having it train against a pool of partners (Lupu et al., 2021; Strouse et al., 2021). Lupu et al. uses an auxiliary loss to induce diversity, while Strouse et al. selects older checkpoints of the model; both suffer from the *arbitrariness* and *sabotage* issues which we address in our work. Canaan et al. (2019) generates a diverse pool of agents through MAP-Elites (Mouret & Clune, 2015), but the algorithm relies on low-dimensional rule based agents and fails to produce strong Hanabi agents (scores $> 19$ points). In contrast, our method scales to high-dimensional settings and does not require manual hard-coding.

Hu et al. (2020) introduce a setting where the goal is to maximize the cross-play (mixed team) performance between independently trained agents from the same training algorithm. For clarity, we refer to this metric as the *intra-algorithm cross-play* (intra-AXP) in this paper. Hu et al. (2020) highlighted the importance of "reasonable" policies, in particular focusing on symmetry breaking. Ma et al. (2022) also investigates producing reasonable policies through the inductive biases of various model architectures. They find that certain architectures achieve diversity in simple settings.

The work most similar to ours is that of Charakorn et al. (2023). They learn incompatible policies by maximizing SP scores, while minimizing XP scores in a pool, and optimizing the lower bound on variations between policies. As we show in Section 7, this approach leads to agents that obtain low XP scores by *sabotaging* the game rather than discovering fundamentally incompatible conventions.

## 3 BACKGROUND

### 3.1 TURN-BASED DEC-POMDPS WITH PUBLIC ACTIONS

In this work, we assume a turn-based Dec-POMDP (Oliehoek, 2012) which can be described by a tuple $(n, \mathcal{S}, \mathcal{A}, P, r, \mathcal{O}, \gamma)$, with number of agents $n$, state space $\mathcal{S}$, action space $\mathcal{A}$, transition function $P : \mathcal{S} \times \mathcal{A} \times \mathcal{S} \rightarrow [0, 1]$ determining the probability over next states, reward function $r : \mathcal{S} \times \mathcal{A} \rightarrow \mathbb{R}$, observations function $\mathcal{O}(s) = o$, and discount factor $\gamma \in [0, 1]$. We also define the trajectory up to time $t$ as $\tau_t = (o_0, a_0, \ldots, a_{t-1}, o_t)$ and the *action-observation history* (AOH) of agent $i$ as $\tau_t^i = (o_0^i, a_0, \ldots, a_{t-1}, o_t^i)$. For a given player $i$, a policy $\pi^i(a|\tau_t^i)$ is a function that maps an AOH to a distribution over actions. Importantly, this setting is turn based, meaning only one agent acts at any given time step. We also assume public actions, such that all agents observe the action selected by the acting agent—a limitation we inherit from OBL.

Multiple trajectories can produce the same AOH. The relationship between them is therefore probabilistic and depends on the policy $\pi$ that generated the trajectory up to time $t$. We refer to this distribution over trajectories as the *belief*, $\mathcal{B}_\pi(\tau_t|\tau_t^i) = P(\tau_t|\tau_t^i, \pi)$.

### 3.2 SELF-PLAY TRAINING

Self-play (SP) is a general class of training methods for multi-agent RL. When training the (joint) policy $\pi$ in SP, we unroll the policy up to time step $t$, producing the partial trajectory $\tau_t$. Each agent $i$ then observes $\tau_t^i$ and samples an action from $\pi(\cdot|\tau_t^i)$. The team receives a reward $r_t = r(\tau_t, a_t)$ and transitions to trajectory $\tau_{t+1}$. In two-player turn-based games, the process is repeated for agent $-i$ to obtain $r_{t+1}$ and $\tau_{t+2}^i$. Finally, SP computes the TD target

$$\delta_{\text{SP}} = r_t + \gamma r_{t+1} + \gamma^2 V_\pi(\tau_{t+2}^i, a), \tag{1}$$

which assumes that actions will be selected according to policy $\pi$ at all future time-steps.

During SP training, past, present and future actions are assumed to be sampled from the policy being trained. In fully cooperative settings, this enables agents to communicate additional information about their private observation (and therefore the trajectory $\tau_t$) by selecting actions to sharpen their partner's belief $\mathcal{B}_{\pi^{-i}}(\tau_t|\tau_t^i)$. Arbitrary correlations arising during training (e.g. due to random initialization) are therefore reinforced, as they provide useful information to other agents at future time-steps. While such correlations can improve the team's return, they are unlikely to reoccur on independent training runs, resulting in a policy that is brittle and difficult to cooperate with.

### 3.3 OFF-BELIEF LEARNING

Off-Belief Learning (Hu et al., 2021b, OBL) is a training algorithm designed to address the shortcomings of SP training in turn-based Dec-POMDPs with public actions. It prevents agents from learning arbitrary and brittle conventions peculiar to the random correlations of a particular run.

The general issue in applying SP training to cooperative MARL is that it enables information feedback loops that reinforce spurious correlations between actions and meanings. For example, consider in the cooperative card game Hanabi a policy $\pi_A$, where by chance $\pi_A$ often takes a particular action X when its partner has a playable card, and otherwise takes action Y. Upon observing X, the partner's belief over the card will be that it is playable. The partner therefore plays it, resulting in a positive reward which reinforces the convention that "X means playable". Crucially, this occurs even if neither X nor Y reveal any extra information about the playable card.

The key insight of OBL is to break this loop by fixing a belief model $\mathcal{B}_0$, which is independent of $\pi_A$. At each step, OBL then reinterprets the AOH based on this $\mathcal{B}_0$ by sampling a new fictitious trajectory $\tau_t' \sim \mathcal{B}_0$. The correlation between the trajectory seen by $\pi_A$ and its action are then restricted to what "survives" given this trajectory resampling. Thus, if $\pi_A$ takes action X when a card is playable, but that same card isn't playable under $\tau_t'$, then no reward is obtained if the partner plays the card. In particular, OBL uses $\mathcal{B}_0 = \mathcal{B}_{\pi_0}$, where $\pi_0$ is fully random. Thus, in expectation, "X means playable" will only be reinforced if X carries verifiable information about the playability of the card.

More formally: like SP, when training policy $\pi$, OBL unrolls $\pi$ to obtain the trajectory $\tau_t$, AOH $\tau_t^i$ and action $a_t^i \sim \pi(\cdot|\tau_t^i)$ for agent $i$. However, OBL breaks the information feedback loop by entirely ignoring the reward $r(\tau_t, a_t^i)$. Instead, it assumes access to the environment simulator, as well as to a belief model $\mathcal{B}_{\pi_0}$ of another policy $\pi_0$, and samples a *fictitious* trajectory $\tau_t' \sim \mathcal{B}_{\pi_0}(\cdot|\tau_t^i)$. In this fictitious trajectory, OBL first applies the "real" action $a_t^i$ and then fictitious action $a_{t+1}^{-i}{}' \sim \pi(\cdot|\tau_{t+1}')$. It thus receives fictitious rewards $r_t' = r(\tau_t', a_t^i)$ and $r_{t+1}' = r(\tau_{t+1}', a_{t+1}^{-i}{}')$, and fictitious future trajectory $\tau_{t+2}'$ and AOH $\tau_{t+2}^i{}'$. The TD target used for training the policy is therefore

$$\delta_{\text{OBL}} = r_t' + \gamma r_{t+1}' + \gamma^2 V_\pi(\tau_{t+2}^i{}'). \tag{2}$$

In essence, rather than training on the real rewards seen during the transition, OBL samples a fictitious transition at every time step. This fictitious transition reinterprets the AOH of agent $i$ as having been produced by policy $\pi_0$ rather than $\pi$. As a result, actions that sharpen the posterior distribution $\mathcal{B}_\pi(\tau_t|\tau_t^i)$ over trajectories given $\pi$ are no longer reinforced in general. Instead, agents must rely on actions which convey information about the trajectory in spite of this resampling.

As stated earlier, OBL assumes $\pi_0$ to be a fully random policy. As a result, spurious correlations between the real trajectory and the observations do not propagate and agents can only rely on actions that convey verifiable information about the state. We refer to such agents as "grounded."

OBL can be iterated to produce a hierarchy of policies, with each level $\pi_\ell$ being trained on fictitious transitions sampled from a belief model $\mathcal{B}_{\ell-1} = \mathcal{B}_{\pi_{\ell-1}}(\tau_t|\tau_t^i)$. This process was shown to reliably produce policies with similar conventions that are devoid of symmetry breaking and are altogether considered as more "reasonable" partners for coordination. In practice, we use neural networks $\hat{\mathcal{B}}_l$ trained with supervised learning to approximate $\mathcal{B}_l$ to enable fast inference and sampling.

## 4 PROBLEM SETTING AND MOTIVATION

Many works seek to train a large pool of diverse policies from scratch. Instead, given access to a fixed *repulser policy* $\mu$, our goal is to train an *adversary* $\pi$—a new agent with a different play style. We predict that many methods addressing the latter task could theoretically be deployed at larger scale to produce a population of agents. However, this is not the primary focus of our paper.

We aim for the adversary to exhibit *meaningful* diversity from the repulser: to adopt conventions and strategies that differ drastically from those of the repulser, to the extent permitted by the environment. This is in contrast to policies that merely exhibit small differences in action probabilities or state occupancy but otherwise converge to the same high level strategy (Lupu et al., 2021). Furthermore, we seek adversaries that achieve high return and that are reasonable teammates. We further discuss these desiderata in section 4.2.

## 4.1 MOTIVATION

The main motivation to our work is to generate quality partners to serve as a test suite for ad-hoc coordination. While it is often easy to produce a population by varying the training method or hyperparameters, there is no guarantee that such approaches will produce meaningfully diverse policies. Additionally, it is concerning if small training variations do produce diversity, as it suggests that the training algorithms used do not reliably output policies with consistent conventions. Indeed, such policies will fail in the context of intra-AXP, making them poor partners for ad-hoc evaluation.

Instead, with a method that is able to take a small number of reasonable partners and output highly skilled and very distinct adversaries, it becomes possible to generate a partner pool that both covers a more substantial portion of the policy set and is less likely to be populated by poor collaborators. With access to such a method, it will become easier to test the ability of an agent to generalize to unseen and meaningfully diverse partners.

A method for producing adversaries to skilled agents has other possible applications. Previous studies (Hu et al., 2021b; Lupu et al., 2021) have shown that biases in the training method or the environment structure can result in RL agents repeatedly converging to the same equilibrium. Discovering new ways of solving a task is therefore an open problem. Were it solved, it would have applications to software testing, for instance by discovering new ways of breaking a feature or simulating different user behaviours when using a program.

## 4.2 DESIDERATA FOR ADVERSARIES

Before proposing an approach to train adversaries, we first establish how to measure success. Whether two policies are *meaningfully* diverse is environment and task-specific. For instance, two humanoid robots moving their arms differently may be considered irrelevant if the goal is to discover new gait patterns. Similarly, the feasibility of training strong and distinct policies is environment-dependent (e.g. a task may have a unique equilibrium). Therefore, we do not formally define the notion of "meaningful diversity", and instead determine the desiderata of such agents, each understood to be potentially limited by the environment itself.

**Skill level:** Past works indicate that diversity often comes at the cost of performance. While the environment structure may in itself be at cause, this effect can also be attributed to the difficulty of simultaneously optimizing for return and diversity (Parker-Holder et al., 2020). A good method would minimize this drop, and produce adversaries that are as strong as allowed by the setting.

Note that this is not claiming that ad-hoc partners of beginner or intermediate levels are not useful. However, it is usually easy to obtain such partners by selecting earlier checkpoints of a trained policy. Thus, our goal is to produce adversaries near expert level (i.e. as close to SOTA as possible).

**"True" Diversity:** Another core criteria to adversaries is that they adopt distinct strategies from those of their respective repulser policy. The first way to evaluate whether the adversary $\pi$ and its repulser $\mu$ adopted different strategies is by evaluating them in XP, as a mixed team. Low XP can be indicative of distinct and incompatible conventions, but it is not sufficient. Indeed, low XP can also be explained by having brittle policies that fail at the slightest deviation from SP. Even worse, since $\pi$ is a function of $\mu$, it is possible that $\pi$ learns to identify when it is paired with $\mu$, and deliberately performs poorly, i.e. sabotages the game (see Section 7).

While secondary, it is also desirable that the adversary strategies differ in an interpretable way from those of the repulser policy. For instance, an adversary to a robot sports team that is particularly aggressive may instead play much more defensively.

**Reasonableness:** We require adversaries to be "reasonable" or "well-behaved" in an informal sense. First, this means policies that do not sabotage, as explained above. Secondly, we wish to avoid arbitrary conventions or symmetry breaking, since those are unlikely to be recovered even by subsequent runs of the same algorithm, making for very inflexible and uncooperative partners.

While we do not have a problem agnostic means of identifying the failure modes listed above, we do have Hanabi specific metrics as explored in Section 6. Furthermore, as explained in Section 5, our method avoids these failure modes *by design*, building on the desirable properties of OBL.

# 5 ADVERSARIAL DIVERSITY

Given a *repulser policy* $\mu$, how do we train an *adversary policy* $\pi = \mathrm{Adv}(\mu)$ satisfying the criteria above? A simple starting point is to train $\pi$ to maximize SP return while minimizing return when paired with $\mu$. However, as we show in section 7, this approach leads to agents that identify whether they are playing in SP and sabotage if not. Thus, it fails to produce reasonable policies. We therefore introduce ADVERSITY, which overcomes these issues with two key insights.

---

**Algorithm 1** ADVERSITY training at level $\ell$ for *one* data collection and training step. We present the two player case. At timestep $t$, the active player is $i$ and the next player is $-i$.

---

$\triangleright \hat{\mathcal{B}}_{\pi_{\ell-1}}$: belief model from previous ADVERSITY level.
$\triangleright \pi_{\ell}$: new ADVERSITY policy being trained, $Q_\theta$: the $Q$-network that constitutes the policy $\pi_\ell$.
$\triangleright \mu$: repulser policy, $\hat{\mathcal{B}}_\mu$: repulser belief model, $\lambda$: repulsive probability.
**procedure** ADVERSITY($Q_\theta, \pi_\ell, \mu, \hat{\mathcal{B}}_\mu, \mathcal{D}$)
    Initialize training episode $\mathcal{E} = \texttt{EmptyList}$ and sample initial environment $\tau_0 \sim P(\tau_0)$
    **while** $\texttt{NotTerminal}(\tau)$ **do**
        Get observation $\tau_t^i = \mathcal{O}^i(\tau_t)$ and action $a_t^i \sim \pi_\ell(\tau_t^i)$ for the active player $i$
        **if** $x \sim \mathcal{U}(0,1); x < \lambda$ **then**
            $\mathcal{B} = \hat{\mathcal{B}}_\mu, \pi_{\text{partner}} = \mu, w = -1$
        **else**
            $\mathcal{B} = \hat{\mathcal{B}}_{\ell-1}, \pi_{\text{partner}} = \pi_\ell, w = 1$
        **end if**
        Sample fictitious trajectory $\tau_t' \sim \mathcal{B}(\cdot | \tau_t^i)$
        Apply active player's action on the *fictitious* trajectory $\tau_{t+1}' = P(\cdot | \tau_t', a_t^i)$ and collect reward $r_t'$
        Partner observes and picks *fictitious* action $a_{t+1}^{-i}{}' = \pi_{\text{partner}}(\mathcal{O}^{-i}(\tau_{t+1}'))$
        Apply partner's action on the *fictitious* trajectory $\tau_{t+2}' = P(\cdot | \tau_{t+1}', a_{t+1}'^{-i})$ and collect reward $r_{t+1}'$.
        Compute target $Q_t = wr_t' + w\gamma r_{t+1}' + \gamma^2 V^{\text{diff}}(\mathcal{O}^i(\tau_{t+2}'))$
        Append observation, action and target to training episode $\mathcal{E}.\texttt{append}((\tau_t^i, a_t^i, Q_t))$
        Apply active player's action on the *real* trajectory and get to the next state $\tau_{t+1} \sim P(\cdot | \tau_t, a_t^i)$
    **end while**
    Add training episode to the replay buffer $\mathcal{D}.\texttt{add}(\mathcal{E})$
    Sample training episode from the replay buffer $\mathcal{E}' \sim \mathcal{D}$
    Do gradient descent $\theta = \theta - \alpha \frac{\partial}{\partial \theta} \mathcal{L}(\theta)$, where $\mathcal{L}(\theta) = \frac{1}{2} \sum_{t \in \mathcal{E}'} [Q_\theta(\tau_t^i, a_t^i) - Q_t]^2$
**end procedure**

---

Firstly, we prevent sabotages by re-sampling the partner for every AOH, such that the adversary cannot condition on the partner's identity to choose its action. At every time step, the adversary's partner is the repulser with probability $\lambda$, and itself otherwise. However, doing so directly perturbs the trajectory distribution and impacts learning. For that reason, we leave the trajectory being unrolled intact and perform this partner switching within the fictitious transitions of OBL. Thus, in "repulsive" transitions (when $\pi$ is paired with $\mu$), we not only sample the next action from the repulser, but also use the repulser's belief model to resample the current trajectory. This reduces the likelihood of $\mu$ being off-distribution and reinforces actions that are incompatible with $\mu$. Conceptually, on those transitions we pretend that all prior actions in the episode were taken by the repulser.

Secondly, we prevent *arbitrary conventions* by training a hierarchy of adversary policies, $\pi_\ell$. On vanilla transitions (when the adversary is paired with itself), each $\pi_\ell$ uses a *fixed* belief model from the adversary policy at the level below, $\pi_{\ell-1}$. Like OBL, the lowest level belief is the grounded belief, i.e. the unique belief corresponding to a uniformly random policy. All belief models are trained using the supervised learning procedure described in Hu et al. (2021a).

Technically, we proceed as follows: at a given level $\ell$, ADVERSITY follows a similar training pattern to OBL by unrolling policy $\pi_\ell$ on the *real* trajectory and computing a fictitious transition given $\tau_t' \sim \hat{\mathcal{B}}_{\ell-1}(\tau_{t+1}^{-i}{}')$ and $a_{t+1}^{-i}{}' \sim \pi(\cdot | \tau_{t+1}^{-i}{}')$. However, with probability $\lambda$, the algorithm utilizes a "repulsive" fictitious transition rather than the vanilla OBL one. In that case, we instead sample $\tau_t' \sim \hat{\mathcal{B}}_\mu(\tau_{t+1}^{-i}{}')$ and the partner's action from $\mu$. We then flip the fictitious rewards to obtain the repulsive target

$$\delta_{\text{Adv}} = -r_t' - \gamma r_{t+1}' + \gamma^2 V_\pi^{\text{diff}}(\tau_{t+2}^i{}'). \tag{3}$$

This procedure is summarized in Algorithm 1 and each transition is illustrated in Figure 1.

| repulser | Self-Play Worst Response | | | ADVERSITY | | |
|---|---|---|---|---|---|---|
| | SP | repulser XP | Intra-AXP | SP | repulser XP | Intra-AXP |
| Rank Bot | $23.86 \pm 0.09$ | $0.0 \pm 0.0$ | $0.64 \pm 1.0$ | $24.22 \pm 0.16$ | $1.94 \pm 0.0$ | $24.09 \pm 0.0$ |
| Color Bot | $23.90 \pm 0.14$ | $0.01 \pm 0.0$ | $1.36 \pm 2.0$ | $24.03 \pm 0.13$ | $2.98 \pm 1.0$ | $10.93 \pm 8.0$ |
| Clone Bot | $23.90 \pm 0.13$ | $0.0 \pm 0.0$ | $6.22 \pm 2.0$ | $23.94 \pm 0.16$ | $7.48 \pm 2.0$ | $21.38 \pm 1.0$ |
| OBL | $23.82 \pm 0.1$ | $0.0 \pm 0.0$ | $3.39 \pm 5.0$ | $24.11 \pm 0.07$ | $9.07 \pm 8.0$ | $8.33 \pm 5.0$ |

Table 1: Score table for SPWR and ADVERSITY for 4 repulser policies. Each number is averaged over 3 independent adversaries. Both approaches produce policies with high SP and low repulser XP, but ADVERSITY achieves higher intra-AXP, which demonstrates that the policies are a more principled and reproducible function of the repulser.

Accounting for both transitions, at every time step the policy learns *difference Q-values* that estimate the expected future discounted *reward difference* between vanilla and repulsive transitions:

$$Q_{\pi_\ell}^{\text{diff}}(\tau_t^i, a^i) = (1-\lambda)\mathbb{E}_{\tau_t' \sim \mathcal{B}_{\ell-1}(\tau_t^i), \, a_{t+1}^{-i}{}' \sim \pi_{\pi_\ell}} \{r_t' + \gamma r_{t+1}' + \gamma^2 V_{\pi_\ell}^{\text{diff}}(\tau_{t+2}^i{}')\}$$

$$+ \lambda\mathbb{E}_{\tau_t' \sim \mathcal{B}_\mu(\tau_t^i), \, a_{t+1}^{-i}{}' \sim \mu} \{-r_t' - \gamma r_{t+1}' + \gamma^2 V_{\pi_\ell}^{\text{diff}}(\tau_{t+2}^i{}')\}$$

$$V_{\pi_\ell}^{\text{diff}}(\tau_t^i) = \sum_a \pi_\ell(a|\tau_t^i)Q_{\pi_\ell}^{\text{diff}}(\tau_t^i, a)$$

Breaking it down, the first line corresponds to vanilla OBL. It implies reinterpreting the past as having been produced by a given policy $\pi_{\ell-1}$ and acting according to $\pi_\ell$ ever after, reinforcing only actions that lead to high expected return when interpreting the partner's actions according to $\pi_{\ell-1}$. This prevents feedback loops in SP where the agent can learn spurious beliefs from noise about which past actions correspond to what unobserved trajectories, and then use those beliefs to signal information and reinforce them into arbitrary and brittle conventions.

The second line represents repulsive transitions. Resampling the current trajectory assuming the AOH was produced by $\mu$ results in $a_t^i$ being reinforced if it is incompatible with $\mu$'s conventions and leads to *low* immediate rewards. At the next step, we assume our partner is $\mu$, thus simulating XP between the repulser and the adversary. Here again, negating $r_{t+1}'$ means we reinforce actions that are misinterpreted by $\mu$.

Finally, in both vanilla and repulsive transitions, we maximize expected future discounted *difference reward*, denoted by $V_\ell^{\text{diff}}(\tau_{t+2}^i{}')$. Conceptually, these *difference value functions* assume that at every point in the future the repulser will intervene for one time step with probability $\lambda$, at which point rewards are inverted and the entire past is re-interpreted according to their belief model $\hat{\mathcal{B}}_{\pi_\mu}$.

Overall, this procedure pushes $\pi_\ell$ to select actions that achieve high return under $\hat{\mathcal{B}}_{\pi_{\ell-1}}$, and that are simultaneously incompatible with $\mu$'s conventions.

This section describes training a single policy $\pi_\ell$ on top of $\hat{\mathcal{B}}_{\ell-1}$ and $\mu$. Because the initial belief $\hat{\mathcal{B}}_0$ is restricted to only rely on grounded information, the skill level of $\pi_1$ is limited. Therefore, to improve skill we follow the procedure from Hu et al. (2021b) and iteratively learn higher levels $\pi_\ell$. At each level, we decrease $\lambda$, up to $\lambda = 0$, at which point our training reverts to vanilla OBL. Nonetheless, since each level uses the belief level of the previous level, in our settings the final play style is incompatible to $\mu$, as we show in Section 7. This shows that we truly obtain novel equilibria.

## 6 EXPERIMENTAL SETUP

Here we describe the evaluation setting and baseline. Training details are included in the Appendix.

### 6.1 HANABI

We implement and test our method in Hanabi, a large scale cooperative card game proposed as a challenging benchmark for Dec-POMDP research (Bard et al., 2020). Hanabi is played with 8 hint tokens, 3 life tokens and a deck of 50 cards, each having a rank between 1 and 5 and one of five colors. It is a game for 2-5 players, but we restrict ourselves to the 2-player version.

|           | Rank Bot        | Color Bot       | Clone Bot       | OBL             | Non-repulser    |
|-----------|-----------------|-----------------|-----------------|-----------------|-----------------|
| SPWR      | $2.36 \pm 0.17$ | $1.14 \pm 0.19$ | $2.17 \pm 0.07$ | $2.04 \pm 0.20$ | $1.57 \pm 0.10$ |
| ADVERSITY | $0.06 \pm 0.01$ | $0.10 \pm 0.02$ | $0.05 \pm 0.01$ | $0.09 \pm 0.05$ | $0.05 \pm 0.01$ |

Table 2: Average number of sabotages per game by the row agents playing with the column agents they are trained to be different from. Each pair is evaluated on 1000 games, averaged over 3 seeds of the row agent. The "Non-repulser" column is when SPWR/ADVERSITY of one agent plays with the other three agents. For reference, in self-play, each OBL agent does 0.057 pure sabotages per game for mistakes, risky bets and it almost never loses all 3 lives.

The goal is to form stacks of cards in rank order for each of the five colors. The final score, between 0 and 25, is given by the number of cards successfully stacked by the end of the game. At any given time, player's have five cards in their hands, and can only see their partner's cards. The players must therefore communicate effectively with their partners so that they can act in an informed manner.

One their turn, a player has up to 20 different actions. They can either a) discard one of their cards, b) attempt to play one of their cards, c) hint at all the cards of a chosen color in the partner's hand, or d) hint at all the cards of a given rank in the partner's hand. Discarding replenishes one hint token. Playing a card results in it being placed on top of the pile of its color if it's the next logical card in that pile. Otherwise, the card is lost and the team loses a life. Hinting provides limited information to the partner and consumes a hint token. Hinting is not allowed if there are no hint tokens left, and discarding is not allowed if all 8 tokens are available. If the team loses all 3 lives, the game ends prematurely, with a score of 0. Finally, the game also ends when there are no cards left in the deck.

## 6.2 SELF-PLAY WORST RESPONSE

As a baseline, we implement a "Self-Play Worst Response" (SPWR) – a PPO agent using the same neural network architecture and hyperparameters as above but trained on SP data with probability $1 - \lambda$ and in XP with the repulser with probability $\lambda$. Our SWPR experiments set $\lambda = 0.25$. An entire game is either SP or XP, with no switching within a single game. When in XP, the rewards received are inverted. This is a simpler version of LIPO from Charakorn et al. (2023).

## 7 RESULTS

We evaluate the skill level, diversity, and reasonableness of our method against the SPWR baseline. For both our method and our baseline, we train 3 adversary seeds for each of 4 repulser policies. In addition to vanilla OBL level 5, the repulsers are 3 of the baseline policies inspired by Hu et al. (2021b); namely Rank Bot, which is an Other-Play (Hu et al., 2020) (and therefore color equivariant) policy favouring rank hints, Color Bot, which is a reward-shaped policy favouring color, and Clone Bot, which is a supervised learning bot trained on human data.

We first see in Table 1 that both the SPWR and ADVERSITY agents achieve high SP, corresponding to high skill in Hanabi, and very low XP scores when paired with their respective repulser, showing that both methods produce policies that are incompatible with their repulser. However, ADVERSITY shows a clear advantage over the SPWR in terms of intra-AXP scores, computed between independent adversary seeds. SPWR produces different policies every run, resulting in a very low Intra-AXP score. In contrast, ADVERSITY agents tend to be quite similar, as shown by the lower SP-XP gap. Adversaries to Color Bot and OBL are exceptions and have low Intra-AXP scores.

To evaluate whether the adversaries exhibit meaningful diversity from their repulser, we first look at *conditional action matrices*, presented in appendix A.4. These display the conditional probability of a player's action $a_{t+1}^i$ given the previous action, $a_t^{-i}$. We find that both ADVERSITY and the SPWR exhibit meaningful diversity from their repulsers and between adversaries to different repulsers. For example, when Rank Bot is the repulser policy, ADVERSITY consistently produces adversaries that use color to indicate playable cards. Similarly, while OBL tends to respond to discard actions by also discarding a card, adversaries to OBL learn instead to discard their last card to hint play.

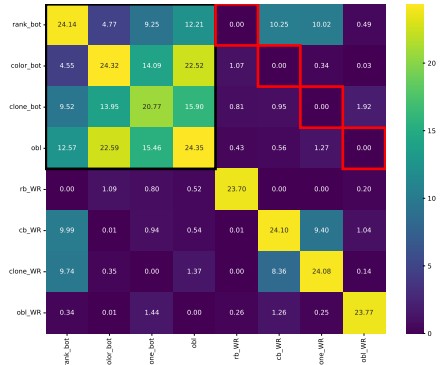 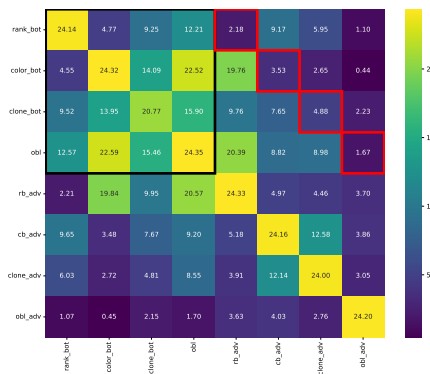

(a) XP matrix with self-play worst response agents      (b) XP matrix with ADVERSITY agents

Figure 2: XP matrices between 8 Hanabi models: 4 repulser models (Rank Bot, Color Bot, Clone Bot, OBL) and their respective adversaries, trained either through SPWR (a) or ADVERSITY (b). Each entry is averaged over 2000 games. Red squares highlight repulser-SPWR or repulser-adversary pairs. Both SPWR and AD-VERSITY produce policies that have low XP scores with their repulsers, but ADVERSITY exhibits a wider range of scores when paired with ad-hoc agents, indicating graceful degradation and more reasonable policies.

In terms of method consistency, the action matrices for different ADVERSITY seeds tend to be similar, reinforcing the idea that it is reproducible. SPWR seeds, on the other hand, differ wildly, explaining the low inter-AXP score mentioned previously.

Finally, Figure 2b shows the XP matrices between bots including the repulsers and one of each adversary seed. Notice that SPWR adversaries have near-zero XP scores with virtually every partner; a red flag supporting the hypothesis that they learned to identify when not in SP and purposely throw the game. Meanwhile, ADVERSITY agents exhibit a graceful degradation of ad-hoc performance depending on the similarity to the partner's policy, indicating much more reasonable policies.

**Sabotaging:** We verify that unlike ADVERSITY, SPWR adversaries exhibit sabotaging behavior in Hanabi. We do this by measuring sabotages, the number of knowingly unplayable cards (based on revealed information) played by the agent. We measure the average number of sabotages per game when SPWR and ADVERSITY are paired with their respective repulser agents and report results in Table 2. ADVERSITY consistently has a low number of sabotages ($< 0.1$) per game, whereas SPWR has at least 1 sabotage per game and in many cases $> 2$. The sabotages are lower for SPWR(Color Bot) simply because a color hint does not immediately reveal whether a card is definitely unplayable. The SPWR(Color Bot) agent simply plays unhinted cards blindly, which is not necessarily a sabotage by our strict definition, but a poor move nonetheless. Moreover, SPWR agents sabotage all non-SP games, not just the ones with the repulser they were trained with. This indicates that the poor XP performance of SPWR comes not from playing a reasonable reward-maximizing strategy that happens to be meaningfully different and incompatible with other agents, but from "deliberately" playing bad actions upon identifying that its current partner is not itself. We also verified that the $< 0.1$ mean incidence of sabotaging for ADVERSITY is in line with vanilla OBL evaluated in SP, i.e. corresponds to a standard rate of mistakes or risky bets. On a different metric, SPWR is responsible for a dominant amount of $2.67 \pm 0.05$ out of 3 life losses per game while, ADVERSITY is only responsible for $1.42 \pm 0.05$—roughly half of the total mistakes. This also indicates that SPWR tries hard to terminate games deliberately, while ADVERSITY policies fail due to meaningful incompatibility, without any party trying to explicitly sabotage.

## 8 CONCLUSION AND FUTURE WORK

In this paper, we introduce ADVERSITY, a method for producing highly skilled and reasonable policies for a fully cooperative task that play according to meaningfully diverse conventions. While our results show that both ADVERSITY and our baseline produce agents that exhibit high skill and meaningful diversity from their repulser, only ADVERSITY agents are also reproducible on independent runs and reasonable, as indicated by the graceful degradation of their performance with different ad-hoc partners.

The main limitation of our method is the high computational cost, making it difficult to scale the method to a large number of adversaries. Were this issue solved, ADVERSITY could theoretically be used to produce a large pool of diverse agents by iteratively computing adversaries to past models.

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

## A  APPENDIX

### A.1  TRAINING DETAILS

Our implementation is based on the open sourced OBL code with two main modifications. We first replace the recurrent Q-learning backbone with PPO as it runs faster and requires significantly less memory. Then, we implement a synchronous method that trains all OBL levels simultaneously.

Otherwise, we use a distributed training set up similar to (Hu et al., 2021b), which we detail in the appendix. All bots are trained for 3000 epochs, each epoch consists of 1000 gradient steps. We select the checkpoint with the highest SP score.

The original OBL trains multiple levels of policies sequentially, using the output policy of the previous level as the input policy of the new level. In ADVERSITY, we train all levels simultaneously for faster wall-clock time. These policies are denoted as $\pi_0, \pi_1 \ldots \pi_L$ and their corresponding belief models are denoted as $\hat{\mathcal{B}}_0, \ldots, \hat{\mathcal{B}}_L$. To warm up the belief model and avoid having too many invalid samples, we first train a belief model $\hat{\mathcal{B}}_0$ on the uniform random base policy $\pi_0$ and initialize all $\hat{\mathcal{B}}_l = \hat{\mathcal{B}}_0$. Then, $L$ policy training tasks and $L$ belief training tasks start at the same time. The belief task of $\hat{\mathcal{B}}_l$ gets a latest copy of $\pi_l$ every 50 gradient steps and the policy task of $\pi_l$ gets a latest copy of $\hat{\mathcal{B}}_{l-1}$ every 50 gradient steps. The details of each individual belief follows the exact configurations of the original OBL paper and each policy task uses the PPO-OBL method described above.

For each adversary, we train a hierarchy of 7 levels, setting $\lambda = 0.25$ for $l = 1$ and decreasing by 0.08 every level (min. 0). Levels $l \leq 4$ are trained simultaneously, followed by levels $l \geq 5$, also trained simultaneously and with beliefs initialized at $\hat{\mathcal{B}}_l = \hat{\mathcal{B}}_4$. This split was forced by limitations on the concurrent compute available to the authors, but we anticipate no change in performance if all levels were trained simultaneously. The ADVERSITY numbers reported in Section 7 all refer to the highest level of the hierarchy.

### A.2  POLICY TRAINING DETAILS

We use a large scale distributed training framework for policy training. To train a single policy, we run 6400 games in parallel, each adding to a centralized replay buffer. We achieve this by running 80 threads in parallel, with 80 games running per thread. All models are on GPUs and we dynamically batch all model calls in order to increase inference speed. This schema also allows games on the same thread to forward environment calls while certain games wait for GPU calls. As done in (Wang et al., 2016), when an environment terminates, each game grabs all necessary objects: observations, actions, and targets, pads everything to a length of 80 and adds it to a centralized replay buffer.

For every training step we apply the PPO update rule, but instead of using the real reward and advantage, we use the fictitious values. Every $m = 10$ training steps, we update the environment actors with the weights for the updated policy. As done in Cui et al. (2021) synchronously train our hierarchy of beliefs and policies, querying for and updating all dependencies every $p = 50$ training steps.

We utilize the same policy architecture as Hu et al. (2021b). We utilize their public-private LSTM architecture. The public observation is encoded by a one-layer feedforwards neural network followed by a LSTM. The private observation is encoded by a three-layer neural network. We combine these encodings via element wise multiplication.

For all OBL experiments we compute the target with $r = 1$ fictitious steps. We also sample the belief model $s = 10$ times and use the first sampled trajectory that doesn't violate card constraints to compute the fictitious targets. We then use a simulator to produce transitions from the valid trajectory. Like Hu et al., we discard the fictitious transition whenever the belief fails to produce a valid sample, which in practice happens on less than 1% of transitions.

**Implementation**

The policy is represented by a public-LSTM network $\pi_\theta$ with a value head and a policy head. A large number of parallel workers generate data by sampling from a slightly outdated policy $\pi_{\theta'}$ and write that data into a replay buffer $\mathcal{D}$. One datapoint in $\mathcal{D}$ is an entire trajectory $\tau^j$. Although PPO

normally does not need a replay buffer, we still use one here to fully decouple inference and training for maximum speed. Its size is set to a small value of 1024 to minimize the instability caused by stale data. $\pi_\theta$ is trained with the Adam optimizer (Kingma & Ba, 2014) on minibatches of data uniformly sampled from the replay buffer. The value loss is $\mathbb{E}_{\tau^i \sim \mathcal{D}} \sum_t [r_t + \gamma V_\theta(\tau_{t+1}^i) - V_\theta(\tau_t^i)]^2$. The policy loss is $\mathbb{E}_{\tau^i \sim \mathcal{D}} \sum_t \min[r_t(\theta)\dot{A}_t, \text{clip}(r_t(\theta), 1 \pm \epsilon)\dot{A}_t]$ where $r_t(\theta) = \frac{\pi_\theta(a_t^i|\tau_t^i)}{\pi_{\theta'}(a_t^i|\tau_t^i)}$, $\dot{A}_t = $ StopGradient$[r_t + \gamma v_\theta(\tau_{t+1}^i) - V_\theta(\tau_t^i)]$. We perform one gradient step per minibatch. We use 1-step bootstrapped value target instead of $\sum_t r_t$ because it converges significantly faster and it fits well in the OBL fictitious target computation. $\pi_{\theta'}$ is synced with $\pi_\theta$ every 10 gradient updates.

### A.3 BELIEF TRAINING DETAILS

We utilize the same distributed training schema from policy training for belief training. This has also been done by (Hu et al., 2021b). As done in policy training, we query and update dependencies every $p = 50$ training steps.

For belief training we store the true hand of the player along with the observation to train the belief. For training, we train an autoregressive belief model that predicts cards oldest to newest via supervised learning. More precisely, the belief model is trained to minimize the loss

$$\mathcal{L}(\mathbf{h}|\tau_t^i) = -\sum_{k=1}^{n} \log p(h_k|\tau_t^i, h_{1:k-1}), \tag{4}$$

where $h_k$ is the $k$th card in the player's hand and $n$ is the hand size (usually 5).

### A.4 ADDITIONAL RESULTS

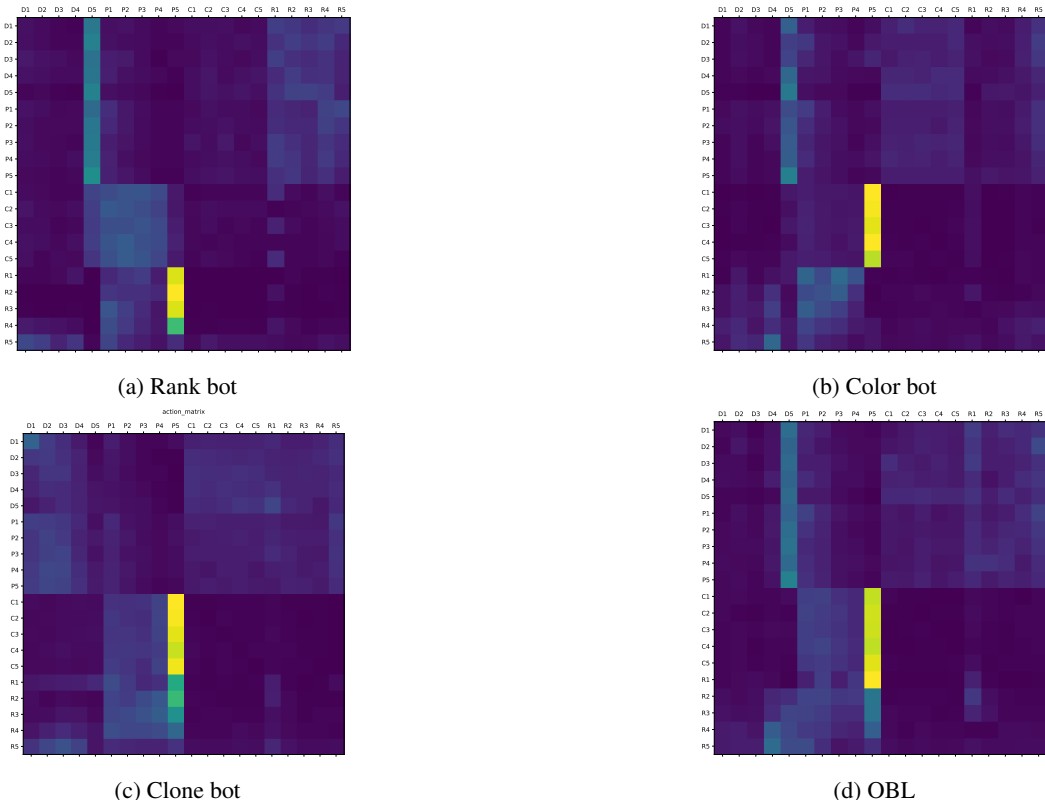

(a) Rank bot

(b) Color bot

(c) Clone bot

(d) OBL

Figure 3: Conditional action matrices showing $p(a_{t+1}|a_t)$ for the 4 repulser policies

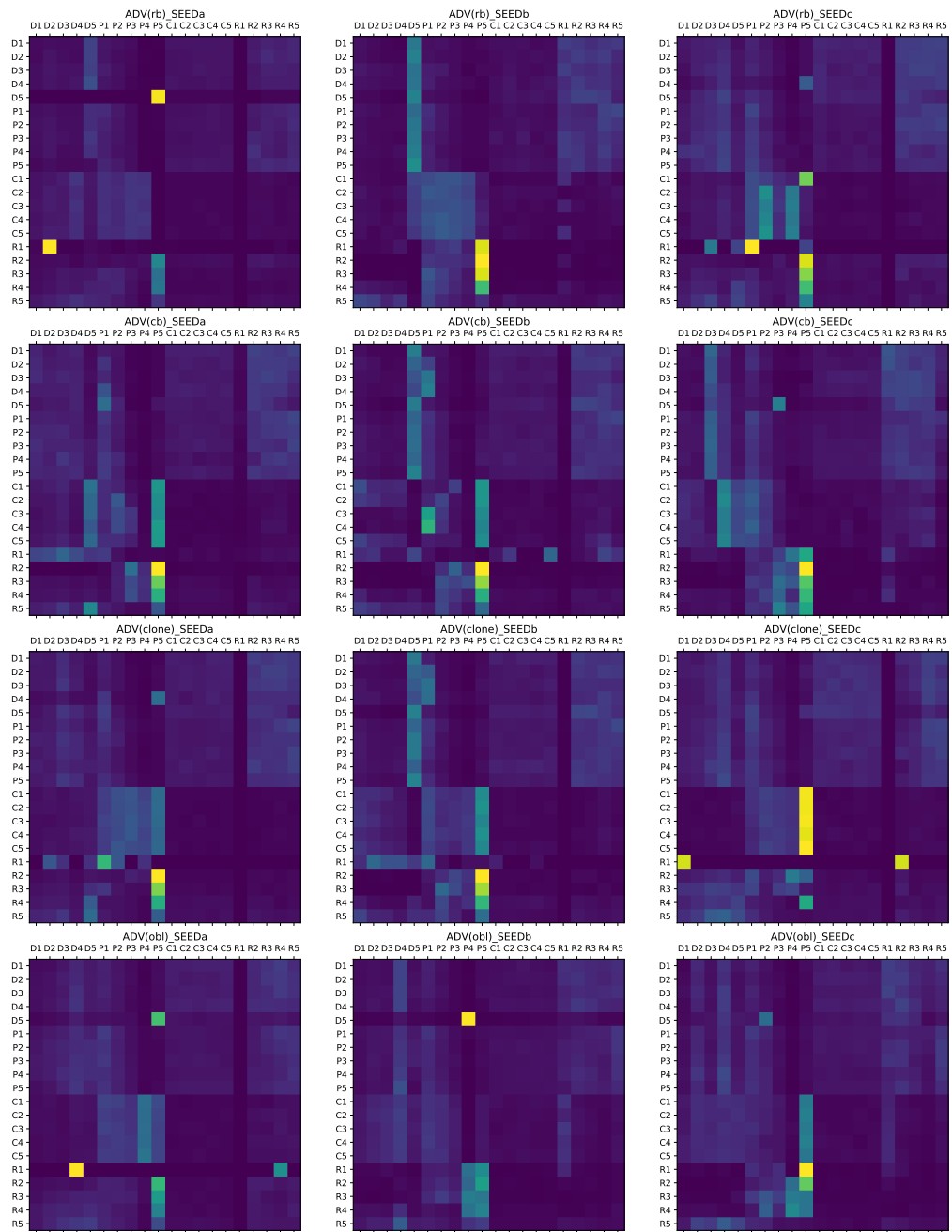

Figure 4: Action matrices for all SPWR agents.

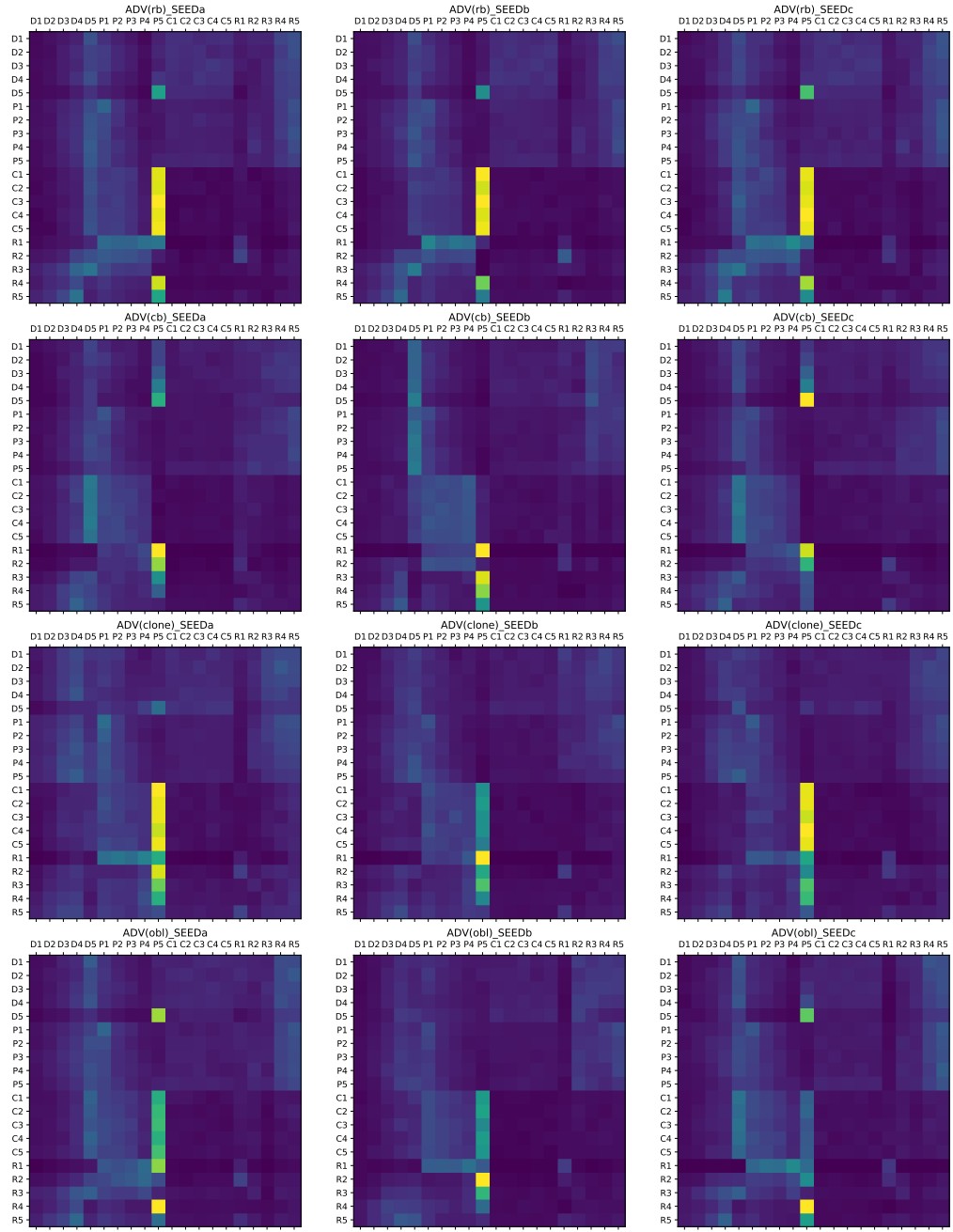

Figure 5: Action matrices for all ADVERSITY agents.

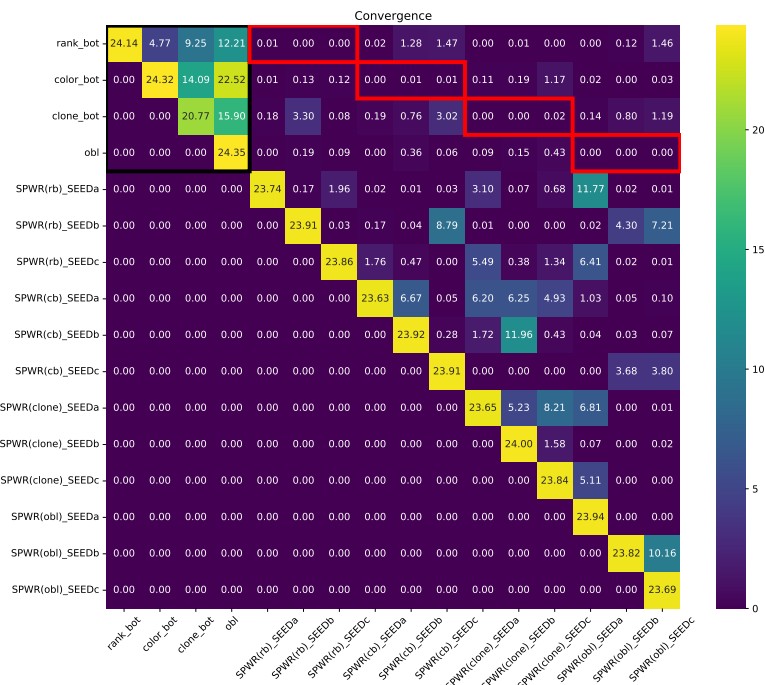

Figure 6: XP matrix of the four repulser candidates and all the SPWR bots. Red rectangles indicate pairs of the form $(X, \mathrm{SPWR}(X))$. Numbers below the diagonal were not computed.

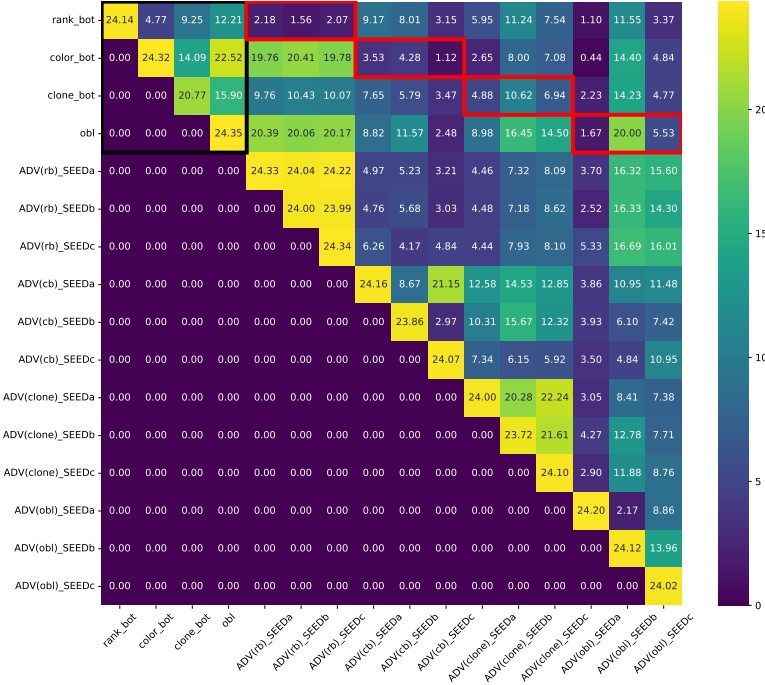

Figure 7: XP matrix of the four repulser candidates and all the ADVERSITY bots. Red rectangles indicate pairs of the form $(X, \mathrm{Adv}(X))$. Numbers below the diagonal were not computed.

