# OpenReview forum: "Adversarial Diversity in Hanabi"
_ICLR.cc/2023/Conference — ICLR 2023 notable top 25%_

### Official Review · Reviewer_awGQ · 2022-10-15

**Confidence:** 4
**Correctness:** 4
**Technical Novelty And Significance:** 2
**Empirical Novelty And Significance:** 2
**Recommendation:** 6

**Clarity, Quality, Novelty And Reproducibility:**

The paper is written clearly, and is easy to follow. Authors did a good job in explaining their motives, method, and results. Regarding the reproducibility, in the abstract it is mentioned that the source code will be released publicly, but it was not shared in the supp material.

**Strength And Weaknesses:**

Tackling the problem of MARL with diverse agents has given little attention in MARL. Authors try to fill this gap in MARL by introducing ADVERSITY which relies on off-belief learning to put more weights on actions with meaningful information about the trajectory.

Questions:
1- How was the SPWR baseline optimized in terms of hyperparameters or lambda? Using the same hyperparameters as ADVERSITY won’t be necessarily the optimal ones for SPWR. And for lambda, why 0.25?

2- According to table 1, the difference between SPWR and ADVERSITY in SP is not statistically significant. Could you discuss this?

3- Results are averaged over 3 seeds which is low. Increasing the number of seeds could be helpful in terms of trusting the results and see how the proposed method is robust against various scenarios.

4- What does “reasonable” mean in the abstract?

**Summary Of The Paper:**

This paper focuses on producing strong and diverse policies in Dec-POMDPs. One important aspect of the proposed method is to prevent the adversary agent from identifying whether it is self-playing or cross-playing by randomizing between the two scenarios. It is also important to bear in mind that in the studied setting in turned based with public actions. The proposed method will lead to having several highly-skilled agents with different styles of play.

**Summary Of The Review:**

See above.

---

> ### Author Response · Authors · 2022-11-10
> **Reply to reviewer awGQ**
>
> We thank reviewer awGQ for their questions, and are glad they found our paper clearly written and easy to follow. They may find our answers below.
>
> **1. Hyperparameters and $\lambda$**
>
> The hyperparameters for ADVERSITY and SPWR are the same, but they were not selected specifically for ADVERSITY. Instead, they were optimized for training a PPO-based SP agent in Hanabi, and were copied over to both ADVERSITY and SPWR. We then set lambda based on trials in ADVERSITY, where we found that $\lambda=0.25$ provided the right balance between diversity, skill, and compute requirements. Indeed, we hypothesize that a higher lambda could work as well if not better given sufficient compute to anneal over more than 4 levels.
>
> The reviewer is indeed correct in saying that optimizing hyperparameters independently for SPWR and ADVERSITY could have produced stronger baselines and results. However, there is no reason to believe it would have solved the pathological sabotaging behavior at the core of our paper. This is because identifying the partner and throwing the game is a local optimum when trying to minimize XP, and one that performs extremely well in light of that objective. The same holds for the low Intra-AXP of SPWR, which has been shown to be a fundamental issue with SP training in cooperative settings that cannot simply be addressed with better optimization [1].
>
> **2. SP scores in Table 1**
>
> As the reviewer points out, and as we comment in the caption of Table 1, both SPWR and ADVERSITY achieve high SP scores. For SPWR, this is to be expected, since 75% of its training data is in SP. What our results show is that ADVERSITY maintains high SP scores and low repulser XP scores while also achieving higher Intra-AXP and avoiding sabotages. Note in general that policies trained to perform well outside SP (e.g. in Intra-AXP) will often inevitably achieve lower SP scores, since many environments admit arbitrarily complex conventions that allow for higher SP scores at the detriment of interpretability and flexibility [1, 2, 3].
>
> **3. Number of seeds**
>
> Because of computational constraints, we were limited in the number of total policies we could train. We thus trained fewer seeds (i.e. 3) on more repulsors (i.e. 4). This allowed us to analyze how our method and baseline perform across a range of different repulsors, which we argue is more informative than simply having more seeds. This is especially true regarding the reviewer's desire to see "how the proposed method is robust against various scenarios".
>
> However, we do want to point out that ADVERSITY results have very small standard errors in Table 2, where we showcase the main benefit of ADVERSITY in that it does not produce policies that deliberately sabotage. Thus 3 seeds suffice to demonstrate a convincing quantitative difference between ADVERSITY and the SPWR baseline. Furthermore, our results are for the most part consistent across different runs. This can for instance be seen in Figure 5, which displays the action matrices for all ADVERSITY agents.
>
> **4. Meaning of "reasonable" in abstract**
>
> Thank you for pointing out that the term "reasonable" is unclear in the abstract. We have now modified the abstract to make it clearer that we consider a policy to be "reasonable" when it is symmetry equivariant, avoids sabotaging behavior and exhibits graceful degradation of performance when paired with ad-hoc partners.
>
> We hope our response addresses the reviewer's questions and comments.
>
> [1] H. Hu, A. Lerer, A. Peysakhovich, and J. Foerster. “Other-play” for zero-shot coordination. In 387 H. D. III and A. Singh, editors, Proceedings of the 37th International Conference on Machine 388 Learning, volume 119 of Proceedings of Machine Learning Research, pages 4399–4410. PMLR, 389 13–18 Jul 2020.
>
> [2] A. Lupu, B. Cui, H. Hu, and J. Foerster. Trajectory diversity for zero-shot coordination. In the International Conference on Machine Learning, pages 7204–7213. PMLR, 2021.
>
> [3] H. Hu, A. Lerer, B. Cui, L. Pineda, D. Wu, N. Brown, and J. N. Foerster. Off-belief learning. 391 ICML, 2021

---

> > ### Comment · Reviewer_awGQ · 2022-11-17
> > **Replying to authords**
> >
> > I would like to thank the authors for taking the time to address my concerns. Their explanations were thorough and sufficient.
> > After reading other reviewers concerns and authors' responses, I intend to keep my recommendation as is.

---

> > > ### Author Response · Authors · 2022-11-17
> > > **2nd Reply to Reviewer awGQ**
> > >
> > > We thank the reviewer for acknowledging our reply and are glad they found our **explanations to be thorough and sufficient**. We also kindly invite the reviewer to consider our recently added "Comment regarding our baseline" at the top of the page.

---

### Official Review · Reviewer_wota · 2022-10-30

**Confidence:** 4
**Correctness:** 4
**Technical Novelty And Significance:** 3
**Empirical Novelty And Significance:** 2
**Recommendation:** 8

**Clarity, Quality, Novelty And Reproducibility:**

The paper is technically sound, though the underlying methods are not particularly novel, as they are a relatively simple modification to the existing OBL algorithm.  What is more novel is the identification of the sabotage issue itself.  There are a few areas where greater detail would be helpful, both for clarity and for reproducibility:

- The process of sampling fictitious trajectories from the belief distribution is not described in sufficient detail.  While this method comes from previous work, it is critical to the algorithm, and so a brief discussion of the details of this process, particularly answering: 1) what is the loss used to train the belief model and 2) does this model require the ability to simulate transitions from arbitrary states?

- More detail on the underlying RL algorithm could be provided. For example, presumably the fictitious transitions are added to a replay buffer, but is there any prioritization done over this buffer?

- Algorithm 1 is a little unclear.  It isn't obvious that the loop is being run over the length of a single trajectory.  The update on the second-to-last line also suggests a tabular Q-update, rather than a gradient descent step.  It might help to split Algorithm 1 into two algorithms, one for the outer RL loop, and another for the fictitious sampling process.

**Strength And Weaknesses:**

The key contribution of this work is the exposition of the issue of partner sabotage when attempting to train mutually incompatible policies as a starting point for ad hoc cooperation.  They also contribute the first method that attempts to address this issue, by leveraging off-belief learning.

The main weakness of this work is the relatively limited experimental evaluation of the proposed method.  While the experiments clearly demonstrate that the ADVERSITY algorithm is able to avoid learning sabotage behavior, they do not evaluate the algorithm as part of a larger stack for learning policies capable of playing Hanabi with a variety of partners.  A more thorough evaluation, where ADVERSITY is used to train a population of different agents, which are then used to train ad hoc policies, might indicate whether the diversity encouraged by the ADVERSITY algorithm is actually helpful for ad hoc play.

**Summary Of The Paper:**

A natural way to train qualitatively different policies for cooperative tasks is to penalize a new joint policy for performing well when individual components of this policy are paired with existing policies.  While this approach can identify multiple "incompatible" solutions to a cooperative task, this incompatibility may be achieved by learning to identify the behavior of existing policies, and then intentionally "sabotage" cooperation.  Experiments conducted in this work demonstrate just this type of behavior in the Hanabi environment.  To address this issue, this work presents the ADVERSITY algorithm, which modifies the existing "off-belief learning" algorithm to train joint policies that are incompatible with an existing policy, but cannot reliably identify that policy in order to sabotage play.

The ADVERSITY algorithm proceeds by rolling out the joint policy being learned.  At each step within this "real" trajectory, rather than computing the reward based on the true trajectory, ADVERSITY (like OBL) samples a "fictitious" trajectory that is consistent with an individual agent's observation history, but is generated by the existing policy that the new policy should fail to cooperate with.  In this way, during training, the new policy for an individual agent can never be sure whether it is interacting with the new joint policy, or whether it is interacting with the existing policy.  It therefore has no incentive to intentionally sabotage cooperation.

Experimental results demonstrate that compared to directly penalizing self-play agents for performing well when paired with existing policies, policies learned with ADVERSITY are far less likely to engage in such sabotage, and are more effective in cross play with other ADVERSITY policies trained with different random seeds.

**Summary Of The Review:**

The main justification for acceptance is the fact that this work is the first to address the problem of "sabotage" behavior when training diverse sets of policies for cooperative tasks.  They clearly demonstrate that their method is able to largely solve the sabotage problem in the Hanabi environment.  That said, the empirical results are limited, and are not sufficient to demonstrate that ADVERSITY can be used to learn "good" strategies for Hanabi in ad hoc play.  Additionally, the ADVERSITY method itself is a relatively straightforward combination of two existing approaches.

---

> ### Author Response · Authors · 2022-11-10
> **Reply to Reviewer wota**
>
> We thank reviewer wota for their valuable feedback and are glad they found our exposition of the sabotage problem novel. Please find our reply to the criticism raised below.
>
> **Experiments**
>
> Regarding our set of experiments, we agree that incorporating ADVERSITY into an ad-hoc coordination stack is a relevant next step, and one of the motivations behind our work.  However, given that diversity generation itself has proved a difficult issue, we feel that addressing this challenge alone is a valuable contribution.
>
> We provide strong evidence that ADVERSITY produces policies which are well suited for building an ad-hoc test set in future work. This evidence includes their diversity, their sabotage-free behavior and the graceful degradation of their XP scores when paired with other independent policies. We contrast this with “Self-play worst-response” (SPWR) policies, which purposely throw the game and as a result obtain near-zero XP scores with most partners. Additionally, as indicated by reviewer awGQ, diversity has been given little attention in cooperative MARL and is relevant by itself, independently of ad-hoc coordination.
>
> **Novelty**
>
> The reviewer states that our "methods are not particularly novel, as they are a relatively simple modification to the existing OBL algorithm". We respectfully disagree since our method introduces a number of novel ideas on top of OBL. Unlike OBL, ADVERSITY performs a stochastic reinterpretation of the past using two different belief models and stochastically selects the partner in the fictitious transition. It also learns using a novel difference value function, which assumes that this stochastic reinterpretation and intervention by the repulsor policy will occur at future steps, recursively. Furthermore, "OBL converges to a unique policy" [1] by design and so the insight that a modification of the OBL fictitious transition mechanism will induce meaningful diversity is far from trivial. Finally, ours is the first method to produce strategically diverse, highly skilled and reasonable policies for Hanabi.
>
> **Belief details**
>
> We admit that the paper could benefit from additional details, which we have now added to section A.3 of the supplemental material. The details are the following:
>
> - The loss we use is the same as in [1] i.e.:
> $\mathcal{L}(\textbf{h}|\tau^i_t) = -\sum_{k=1}^n\textrm{log}~p(h_k|\tau^i_t,h_{1:k-1})$,
> where $h_k$ is the $k$-th card in the player's hand and $n$ is the hand size (usually 5).
>
> - Like OBL, the requirement for our method is to be able to simulate valid states (e.g. states that don’t violate card counts). We use a simulator to produce transitions from the valid states. As in [1], we discard the fictitious transition whenever the belief fails to produce a valid sample, which in practice happens on less than 1% of transitions. It is feasible to relax the assumption of simulator access by substituting the simulator for a world model [2] trained to reproduce the environment dynamics. We believe this to be an interesting direction for future work.
>
> **RL and implementation details**
>
> The details for our underlying RL algorithm and its implementation are in sections A.1 and A.2 of the supplemental material of our original submission. As stated there, we use a replay buffer of size 1024 and we train our policy on "minibatches of data uniformly sampled from the replay buffer", which implies that we use no prioritization scheme.
>
> **Algorithm 1**
>
> We thank the reviewer for their concerns around our description of algorithm 1. We have updated algorithm 1 accordingly in the paper.
>
> Please let us know if you have any additional feedback.
>
> [1] H. Hu, A. Lerer, B. Cui, L. Pineda, D. Wu, N. Brown, and J. N. Foerster. Off-belief learning. 391 ICML, 2021
>
> [2] Ha, D., & Schmidhuber, J. (2018). World models. arXiv preprint arXiv:1803.10122.

---

> > ### Comment · Reviewer_wota · 2022-11-16
> > **Reviewer response**
> >
> > So I would tend to agree that the novelty of the proposed method is not really a weakness for this paper, given that the underlying insights are novel.
> >
> > I still have concerns about the experimental results, however.  The issue is that in restricting the class of policies learned by ADVERSITY in an attempt to avoid sabotage, the resulting policies may not be sufficiently diverse to still be useful for training "robust" strategies for play with human partners.  The potential value of ADVERSITY is the ability to train populations of strategies that are sufficiently diverse, without including unrealistic sabotage behavior.  The risk is that ADVERSITY might end up reducing the "useful" diversity of the population, to the extent that the advantage of adversarial training over simpler approaches is lost altogether.

---

> > > ### Author Response · Authors · 2022-11-17
> > > **2nd Reply to Reviewer wota**
> > >
> > > We thank the reviewer for engaging in the discussion. We invite them to read our follow-up below. We also kindly invite them to read the "Comment regarding our baseline" addressed to all reviewers.
> > >
> > > First, we hope to alleviate the concern that ADVERSITY is too restrictive of the class of policies that can be learned. As stated, ADVERSITY prevents policies from learning sabotages. Without sabotages, policies are forced to discover new ways of becoming incompatible and minimizing XP with the repulsor policy. This means learning incompatible policies that are self-contained and that do not rely on identifying the partner and throwing the game. Instead, policies learn conventions that are meaningfully distinct from those of the repulsor.
> > >
> > > We provide evidence for this in Figure 5 of the Appendix, where we visualize the conventions learned by ADVERSITY policies. Comparing those to the repulsor action matrices in Figure 3, we see that ADVERSITY policies are very different from their respective repulsor, at a level matching or exceeding the diversity observed by SPWR policies (our baseline). In fact, no other method has come close to producing similar levels of strategic diversity in Hanabi.
> > >
> > > Secondly, we respectfully disagree that the main value of ADVERSITY is in generating a train time population. Human-AI coordination is indeed a major goal of cooperative MARL, and population-based approaches may indeed be an important part of solving this task. We therefore do agree that training a best response (BR) policy to a pool of ADVERSITY agents and then evaluating that BR with human partners is a worthwhile experiment for future work.
> > >
> > > However, testing agents with human partners is generally difficult and costly, which is why many works have focused on proxy settings such as ad-hoc coordination. In the literature, ad-hoc coordination is often evaluated on populations obtained by hand-crafting policies [1, 2, 4], varying hyperparameters [3], or deploying different RL algorithms on the same task [1, 2, 3]. Thus, the amount of diversity achieved in the test set is unclear, since it is a byproduct of the variability of the algorithms used rather than something that is actively optimized for. Furthermore, if using naive RL algorithms, the policies obtained are likely to exhibit idiosyncratic behavior such as symmetry breaking, which makes them poor choices as members of a test set. That is because it is difficult to untangle whether low XP scores are due to the idiosyncrasies of the test partner, or to a failing of the policy being evaluated. As a result, there is a need for diverse and standardized test sets on which to evaluate the performance of different coordination methods. Generating such test populations is the main motivation behind our work, as stated in our introduction. In addition to diversity, we focus on preventing sabotages because we wish to produce test-time partners that act "in good faith" and are generally collaborative rather than adversarial.
> > >
> > > We hope this addresses the reviewer's concerns, and if so we ask that they please consider updating their score.
> > >
> > > [1] Zand, J., Parker-Holder, J., & Roberts, S. J. (2022). On-the-fly Strategy Adaptation for ad-hoc Agent Coordination. arXiv preprint arXiv:2203.08015.
> > >
> > > [2] Albrecht, S. V. (2015). Utilising policy types for effective ad hoc coordination in multiagent systems.
> > >
> > > [3] Nekoei, H., Badrinaaraayanan, A., Courville, A., & Chandar, S. (2021, July). Continuous coordination as a realistic scenario for lifelong learning. In International Conference on Machine Learning (pp. 8016-8024). PMLR.
> > >
> > > [4] Barrett, S., Rosenfeld, A., Kraus, S., & Stone, P. (2017). Making friends on the fly: Cooperating with new teammates. Artificial Intelligence, 242, 132-171.

---

### Official Review · Reviewer_p7TH · 2022-11-02

**Confidence:** 2
**Correctness:** 3
**Technical Novelty And Significance:** 3
**Empirical Novelty And Significance:** 3
**Recommendation:** 6

**Clarity, Quality, Novelty And Reproducibility:**

CLARITY: The paper is well written and easy to follow.

QUALITY: The paper is lacking some theoretical evidence as to why the proposed method is supposed to work well in general settings.

NOVELTY: As far as I am concerned, the proposed techniques are novel.

REPRODUCIBILITY: The results presented in the paper are reproducible and the authors plan to make their code accessible open source.

**Strength And Weaknesses:**

STRENGTHS

- The problem studied in the paper is interesting for the community focused on the designed of learning agents in games, as well as for the multi-agent reinforcement learning community.

- The paper is well written and all the results are adequately commented.

WEAKNESSES

- While the paper has a mainly practical focus, I would have liked some theoretical analysis and/or formal claims explaining how the method is supposed to work. The experimental results seem promising in Hanabi, but further evidence is needed to conclude that the method could be applied to other domains.

**Summary Of The Paper:**

The paper studies the problem of computing "diverse" joint policies in Decentralized POMDPs (Dec-POMDPs) with public actions. The paper proposes a method based on off-belief learning augmented with "repulsive" fictitious transitions to encourage diversity. Finnally, the proposed algorithm (ADVERSITY) is run on the card game Hanabi to produce new playing agents with diverse play styles.

**Summary Of The Review:**

Overall, I think this is a good paper on an interesting topic. I only have some concerns on the theoretical groundings of the proposed method.

---

> ### Author Response · Authors · 2022-11-10
> **Reply to p7TH**
>
> We thank reviewer p7TH for their comments and are glad they thought this was a good paper.
>
> **Theoretical analysis**
>
> The ADVERSITY method addresses a fundamental issue of training cooperative MARL policies that are at the same time diverse, skilled and that avoid identifying partners and sabotaging games. The latter in particular is a pathological issue demonstrated by our SPWR baseline.
> However, the method is far from straightforward and involves a number of novel ideas, including stochastic reinterpretation of the past, stochastic partner choice and learning with a “difference value function” that assumes a randomly branching process at all future time steps. Unfortunately, this makes the theoretical analysis of our method rather challenging. For example, we have been trying (and failing!) to formulate an objective function which ADVERSITY optimizes. Nonetheless, we hope to provide theoretical analysis of our method and of diversity in cooperative MARL in general, in future work.
>
> **Applicability to other domains**
>
> We would also like to raise a few points that we hope will address the reviewer’s concerns about ADVERSITY being applicable to other domains. First, there is ample precedent in the cooperation literature for papers using Hanabi as the sole complex environment on which they report results [1, 2, 3, 4, 5, 6]. This is in great part because of a lack of other suitable benchmarks for coordination. Indeed, Hanabi is one of the few available Dec-POMPDs to require such a high level of coordination to perform well and to also admit a very broad range of different strategies. This is why it was proposed as a coordination challenge [7] and why we considered it to be a prime choice to demonstrate the efficacy of our method.
>
> Lastly, “sabotages” arise whenever the trained adversary learns to identify its partner and subsequently “throw the game” (i.e. take catastrophic actions). This makes it a general pathology that isn't limited to Hanabi but instead can occur in many other Dec-POMDPs. ADVERSITY prevents sabotages by design and as such would be applicable beyond Hanabi, to produce diverse and skilled agents in other fully-cooperative environments.
>
> We encourage p7TH to share if they have any additional suggestions on improving the paper.
>
> [1] A. Lupu, B. Cui, H. Hu, and J. Foerster. Trajectory diversity for zero-shot coordination. In International Conference on Machine Learning, pages 7204–7213. PMLR, 2021.
>
> [2] Siu, H. C., Peña, J., Chen, E., Zhou, Y., Lopez, V., Palko, K., ... & Allen, R. (2021). Evaluation of human-AI teams for learned and rule-based agents in Hanabi. Advances in Neural Information Processing Systems, 34, 16183-16195.
>
> [3] H. Hu, A. Lerer, A. Peysakhovich, and J. Foerster. “Other-play” for zero-shot coordination. In H. D. III and A. Singh, editors, Proceedings of the 37th International Conference on Machine Learning, volume 119 of Proceedings of Machine Learning Research, pages 4399–4410. PMLR, 13–18 Jul 2020.
>
> [4] H. Hu, A. Lerer, B. Cui, L. Pineda, D. Wu, N. Brown, and J. N. Foerster. Off-belief learning. ICML, 2021. URL https://arxiv.org/abs/2103.04000.
>
> [5] Nekoei, H., Badrinaaraayanan, A., Courville, A., & Chandar, S. (2021, July). Continuous coordination as a realistic scenario for lifelong learning. In International Conference on Machine Learning (pp. 8016-8024). PMLR.
>
> [6] Canaan, R., Togelius, J., Nealen, A., & Menzel, S. (2019, August). Diverse agents for ad-hoc cooperation in hanabi. In 2019 IEEE Conference on Games (CoG) (pp. 1-8). IEEE.
>
> [7] Bard, N., Foerster, J. N., Chandar, S., Burch, N., Lanctot, M., Song, H. F., ... & Bowling, M. (2020). The Hanabi challenge: A new frontier for AI research. Artificial Intelligence, 280, 103216.

---

> > ### Author Response · Authors · 2022-11-17
> > **Following up with reviewer p7TH**
> >
> > We again thank the reviewer for taking the time to review our paper. We are following up to see if they had any more questions or feedback. We also invite them to consider the recently added "Comment regarding our baseline" at the top of the page.
> >
> > Additionally, to summarize our last response:
> >
> > 1. Our method addresses a fundamental issue in cooperative MARL by training policies that are at once diverse, highly skilled, and that avoid sabotages. Unfortunately, its complexity makes theoretical analysis particularly challenging.
> >
> > 2. Like many other papers in the field, we focus our experimental efforts on the card game Hanabi. This is because there are few available settings that present such a rich and complex coordination challenge, which is why the game was proposed as a frontier for AI research.
> >
> > 3. Nonetheless, sabotages are a general pathology and ADVERSITY avoids them **by design**. This makes ADVERSITY applicable to other cooperative domains beyond Hanabi.
> >
> > Please let us know if you have additional questions or comments.
> >
> > Kind regards,
> >
> > The Authors of Paper 3595

---

### Author Response · Authors · 2022-11-17
**Comment regarding our baseline**

We would like to thank the reviewers for their valuable feedback and for engaging in further discussion following our reply.

It has come to our attention that another paper, "_Generating Diverse Cooperative Agents by Learning Incompatible Policies_" [1], is currently under review at ICLR with scores (8, 8, 8, 8). The main method of this paper, “Learning Incompatible Policies” (LIPO), is essentially our naive Self-Play Worst Response (SPWR) baseline. In our paper, we demonstrate that policies trained with SPWR suffer from the issue of “sabotage”, where they will learn to identify their partner and deliberately throw the game if not playing in SP (rather than learning meaningfully diverse policies). We also propose ADVERSITY, a method specifically designed to prevent sabotages while producing policies with diversity and skill levels comparable or exceeding those obtained by SPWR.

Given that we not only identify a fundamental issue with the LIPO paper but also propose and evaluate a method that addresses this, we feel that our scores seem poorly calibrated in comparison.

We provide a more detailed analysis and comparison below.

**Additional Details**

Like us, the authors of LIPO find that it produces behavioral diversity (albeit in environments that are much simpler than Hanabi), but do not probe their policies for sabotage. However, with the knowledge of our own work, we believe their results contain telltale signs of sabotaging. Our reasoning is the following:

In section 4.6, the authors evaluate LIPO and other diversity methods in a multi-recipe variant of Overcooked. In this variant, agents must complete any of the 6 available recipes, at which point the episode terminates and they receive a reward.

For each diversity method $M$, the authors train a population of policies $P_M$, and then a "generalist" best response policy $\pi_M = BR(P_M)$ that is trained to perform well with all the agents in the given population. They evaluate these generalist policies with held-out populations produced by each of the diversity methods. Crucially, they also evaluate the generalists with a population of "Specialists", which are SP policies reward-shaped to only learn to cook a specific recipe. As the authors state, "the specialist population is created for evaluating the agent when the partner has a strong preference". In other words, they are difficult partners because they have been trained to ignore all recipes other than the one they were optimized for.

In Fig. 9 of the paper, we observe that generalists trained on the baseline populations achieve at least a 87% success rate with other baseline populations, a 58.5% mean success rate with Specialists, and a 46.5% mean success rate with the LIPO population. The generalist policy $\pi_{LIPO}$ achieves only 67% success rate with the held-out LIPO population, despite being trained with a LIPO population.
This indicates that the held-out LIPO population is a more difficult test set than even the Specialist policies, which were specifically trained to ignore all attempts at cooking 5 out of 6 recipes. While we don't have access to the LIPO models to verify this, this result is consistent with LIPO policies learning to sabotage in Overcooked. That is because it is more difficult to achieve high returns with partners that deliberately “throw the game” than with partners that merely have strong preferences.

[1] https://openreview.net/forum?id=UkU05GOH7_6

---

> ### Comment · Reviewer_wota · 2022-11-18
> **Reviewer Response**
>
> So I think part of the concern is that while this work appears improves upon the LIPO baseline in Hanabi, the experimental evaluations aren't as thorough, since no generalist policies are trained using the population, and experiments are only done in Hanabi.  It isn't clear that results would generalize to other domains (like Overcooked) where some ability to recognize and adapt to the partner's strategy might be essential for cooperation.  The LIPO work also compares against a wider variety of baseline diversity approaches.  This work also presents LIPO as an existing baseline, and seems to position ADVERSITY as an improvement over LIPO.  From that perspective this work is not as novel as the original LIPO work on which it appears to be based.

---

> > ### Author Response · Authors · 2022-11-18
> > **Response**
> >
> > We wish to thank reviewer wota for being actively engaged in the review process and for providing useful feedback.
> >
> > **Novelty**
> >
> > LIPO was initially presented at the ICML 2022 AI4ABM Workshop on July 23rd, 2022. This makes their work concurrent with ours. As a result, we did not build on LIPO, but rather formulated SPWR as a naive baseline that simply maximizes SP while minimizing XP. We did so independently to demonstrate the problem of sabotaging and then happened to find the LIPO workshop paper, which presents a closely related algorithm as their main method.
> >
> > **Environment**
> >
> > Our evaluation is performed on Hanabi, a well-known and particularly challenging benchmark for coordination. We argue that this is preferable to only evaluating our method on  hand-crafted environments for the purpose of the paper. In Appendix G, the LIPO paper admits that the selected environments are one of the limitations of their work because they use a hand designed variant of Overcooked, making comparisons to existing work challenging.
> >
> > Hanabi is also a significantly more complex environment than Overcooked, requiring much tighter coordination. The "_ability to recognize and adapt to the partner's strategy_" **is** essential in Hanabi, and is one of the key aspects that make it an important benchmark. The Hanabi challenge paper states as a selling point that "_good strategies are not unique, and a robust player must learn to recognize intent in other agents’ actions and adapt to a wide range of possible strategies_." [1] It follows that algorithmic advances made in Hanabi will more readily apply to Overcooked or other domains than the other way around.
> >
> > **Baselines**
> >
> > We reiterate that our goal is to train strong and diverse policies that do not exhibit idiosyncrasies or sabotaging behavior. This is in the scope of building a standardized set of novel test partners to be used in future coordination research. With this scope in mind, we are not aware of any suitable diversity method to be used as a baseline. The LIPO authors compare their method with Multi-SP, Multi-SP+MI, MAVEN and TrajeDI, which suffer from the following issues:
> >
> > - Multi-SP and Multi-SP+MI are just independent runs of SP training, with or without a mutual information bonus. We already know from past works that SP-based methods are poor ad-hoc partners because they rely on arbitrary symmetry breaking and fail to collaborate even with teammates that vary in nothing but the training seed [2, 3].
> >
> > - MAVEN is an algorithm developed for the setting of centralized control and decentralized execution (CTDE). It conditions on a variable $z$ and "_can then be seen as exploring the space of joint behaviours_" [4]. It therefore suffers from the same pathologies as SP training.
> >
> > - TrajeDi is a population-based training (PBT) method that enhances a population of agents trained in SP with a centralized information theoretic diversity loss [5]. Lupu et al. use it to stabilize and robustify a best response policy in Hanabi, but it does not induce meaningful behavioral diversity. They remark so in appendix C.1, stating that "_diversity heatmaps between pools trained with and without TrajeDi have only very subtle differences_". TrajeDi also relies on SP at its core and therefore can also converge to idiosyncratic policies.
> >
> > All of these methods are in essence based on SP and as such are ill-suited for training ad-hoc test partners.
> >
> > **Generalist**
> >
> > Our paper begins to address the problem of generating a meaningful pool of test-time partners to evaluate different ad-hoc coordination methods. Without access to a meaningful test pool, the problem of training a "generalist" policy is ill-defined since we do not know what it will be evaluated on, what it will have to _generalize to_, or how success will be measured. Therefore, without a clearly formalized problem setting, (e.g. the distribution of test agents), training a best response to a population of ADVERSITY agents (or any other population) can provide little to no insights about that population.
> >
> > [1] Bard, N., Foerster, J. N., Chandar, S., Burch, N., Lanctot, M., Song, H. F., ... & Bowling, M. (2020). The hanabi challenge: A new frontier for ai research. Artificial Intelligence, 280, 103216.
> >
> > [2] H. Hu, A. Lerer, A. Peysakhovich, and J. Foerster. “Other-play” for zero-shot coordination. In H. D. III and A. Singh, editors, Proceedings of the 37th International Conference on Machine Learning, volume 119 of Proceedings of Machine Learning Research, pages 4399–4410. 13–18 Jul 2020.
> >
> > [3] H. Hu, A. Lerer, B. Cui, L. Pineda, D. Wu, N. Brown, and J. N. Foerster. Off-belief learning. ICML, 2021. URL https://arxiv.org/abs/2103.04000.
> >
> > [4] Mahajan, A., et al. (2019). Maven: Multi-agent variational exploration. Advances in Neural Information Processing Systems, 32.
> >
> > [5] Lupu, A., Cui, B., Hu, H., & Foerster, J. Trajectory diversity for zero-shot coordination. In International Conference on Machine Learning (pp. 7204-7213). 2021.

---

> > > ### Comment · Reviewer_wota · 2022-11-18
> > > **Reviewer response**
> > >
> > > So the fact that this was concurrent work with LIPO does change things a little, as it significantly increases the novelty of the work.  I still believe there are some limitations regarding the experimental evaluation of ADVERSITY, but weighing these against the greater apparent novelty of the method, I do feel that a higher score is justified, and have adjusted my review accordingly.

---

### Decision · Program_Chairs · 2023-01-20

**Decision:**

Accept: notable-top-25%

**Justification For Why Not Higher Score:**

The lack of theoretical analysis and an empirical analysis limited to one domain make this paper not deserve an oral presentation.

**Justification For Why Not Lower Score:**

The paper contains significant results that can be of interest to the ICLR audience.
However, I am fine also with a poster presentation.

**Metareview: Summary, Strengths And Weaknesses:**

The paper proposes an approach to compute strong and diverse joint policies in a Decentralized MDP. The proposed algorithm is then tested in the Hanabi card game to produce agents with diverse policies.
The problem faced in the paper is relevant and the paper is well-written, easy to follow, and technically sound.
The reviewers raised some issues about the lack of theoretical analysis, a limited empirical analysis, and a limited novelty.
However, after reading each others' reviews and the authors' feedback, the reviewers solved most of their concerns and agreed that this paper deserves publication.
The authors need to implement the suggestions provided by the reviewers while preparing the camera ready.

**Note From Pc:**

if the above contains the word "oral" or "spotlight" please see: "oral" presentation means -> notable-top-5% and "spotlight" means -> notable-top-25%. As stated in our emails, we are disassociating presentation type from AC recommendations

**Summary Of Ac-Reviewer Meeting:**

N/A